# Manual Therapy Improves Fibromyalgia Symptoms by Downregulating *SIK1*

**DOI:** 10.3390/ijms25179523

**Published:** 2024-09-01

**Authors:** Javier Bonastre-Férez, Karen Giménez-Orenga, Francisco Javier Falaguera-Vera, María Garcia-Escudero, Elisa Oltra

**Affiliations:** 1Escuela de Doctorado, Universidad Católica de Valencia San Vicente Mártir, 46001 Valencia, Spain; javier.bonastre@mail.ucv.es (J.B.-F.); karen.gimenez@mail.ucv.es (K.G.-O.); 2School of Health Sciences, Universidad Católica de Valencia San Vicente Mártir, 46001 Valencia, Spain; fj.falaguera@ucv.es (F.J.F.-V.); maria.escudero@ucv.es (M.G.-E.); 3Department of Pathology, School of Medicine and Health Sciences, Universidad Católica de Valencia San Vicente Mártir, 46001 Valencia, Spain

**Keywords:** fibromyalgia, myalgic encephalomyelitis/chronic fatigue syndrome (ME/CFS), pressure point threshold (PPT), physiotherapy, manual therapy (MT), NCT04174300, *SIK1*

## Abstract

Fibromyalgia (FM), classified by ICD-11 with code MG30.0, is a chronic debilitating disease characterized by widespread pain, fatigue, cognitive impairment, sleep, and intestinal alterations, among others. FM affects a large proportion of the worldwide population, with increased prevalence among women. The lack of understanding of its etiology and pathophysiology hampers the development of effective treatments. Our group had developed a manual therapy (MT) pressure-controlled custom manual protocol on FM showing hyperalgesia/allodynia, fatigue, and patient’s quality of life benefits in a cohort of 38 FM cases (NCT04174300). With the aim of understanding the therapeutic molecular mechanisms triggered by MT, this study interrogated Peripheral Blood Mononuclear Cell (PBMC) transcriptomes from FM participants in this clinical trial using whole RNA sequencing (RNAseq) and reverse transcription followed by quantitative Polymerase Chain Reaction (RT-qPCR) technologies. The results show that the salt-induced kinase *SIK1* gene was consistently downregulated by MT in FM, correlating with improvement of patient symptoms. In addition, this study compared the findings in a non-FM control cohort subjected to the same MT protocol, evidencing that those changes in *SIK1* expression with MT only occurred in individuals with FM. This positions *SIK1* as a potential biomarker to monitor response to MT and as a therapeutic target of FM, which will be further explored by continuation studies.

## 1. Introduction

The latest version of the International Classification of Diseases (ICD-11) adopted by the WHO (World Health Organization) in May 2019 [1] classifies fibromyalgia (FM) as a multifactorial chronic primary widespread pain syndrome (code MG30.0) presenting diffuse pain in at least four of the five body regions, anxiety, depression, and overall functional disability [1].

Diagnosis is based on clinical criteria defined by the ACR (American College of Rheumatology) 1990 case definition with revisions [2,3]. Appropriate diagnosis should ensure that pain is not directly attributable to a nociceptive process but consistent with nociplastic pain [1], which is caused by poorly understood mechanisms, rather than local nerve damage (neuropathic pain) [4]. Additional symptoms include non-restorative sleep, fatigue, cognitive impairment, and intestinal problems, overlapping with symptoms present in Myalgic Encephalomyelitis/Chronic Fatigue Syndrome (ME/CFS) [5,6]. Presentation of FM peaks between 20 and 55 years with marked an increased prevalence in women [7,8]. Epidemiology reports vary across countries and regions, with worldwide impact showing 2.7% of the general population and 3.7% in the Valencian Community of Spain studied here [7,8,9].

Because FM etiology and pathophysiology remain unknown, current treatments are directed to palliate symptoms, often leading to polypharmacy [10] and further health deterioration. Clinical guidelines on non-pharmacological therapies include passive therapies, such as hyperbaric oxygen therapy, repetitive transcranial magnetic stimulation, and others, including manual therapy (MT). Positive effects of physiotherapy (e.g., MT) on pain, physical capacity, and quality of life have been repeatedly reported [11]. Previously described effects of pressure therapeutics point at medium load pressure massage (4.5 N) maneuvers, including frequency and repetitions, to aid in muscle deconditioning regrowth, which is typically lost during immobilization or in sedentary individuals, such as severely affected FM and/or CFS/ME patients [12], as described by Dupont–Vergesteegden’s group [13]. At the same time, and similar to CBT (Cognitive Behavioral Therapy) and mindfulness, MT might engage the patient’s mind into relaxation, boost happiness, and promote immune, hormonal, and neurotransmitter responses [14,15].

Tissue reconditioning and patient symptom improvement are always preceded by molecular changes, which often systemically spread through the body fluids, providing a low-invasive opportunity to understand and monitor patient health status through gene expression changes in the blood of patients. In particular, gene expression studies of FM PBMCs (Peripheral Blood Mononuclear Cells) have detected molecular differences coinciding with immune cell activation [16], including hyperactivation of NK cells [17], positing the relevance of immunomodulatory therapeutics, such as MT.

With the intention of elevating our knowledge of MT-mediated healthcare effects on FM, our group developed an MT pressure-controlled custom protocol for FM, resulting in hyperalgesia/allodynia, fatigue, and patient’s quality of life benefits in a cohort of 38 FM cases [18]. The registered clinical trial (NCT04174300) also built a biobank collection of blood samples taken at different points of the treatment, which were analyzed in this study to help understand the molecular mechanisms behind patients’ response to MT. This was mainly achieved by studying the potential correlations between differential gene expression (DE) in patient and control PBMCs with MT and their potential correlation with changes in symptoms. DE was assessed by whole RNA sequencing (RNAseq) and reverse transcription followed by quantitative Polymerase Chain Reaction (RT-qPCR) technologies, and changes in symptoms were measured with the following validated instruments: the FIQ (fibromyalgia impact questionnaire) [19,20], MFI (multi-fatigue inventory) [21], and SF-36 [22] to assess pain, fatigue, and quality of life, respectively.

## 2. Results

### 2.1. Study Design, Demographics, and Phenotyping

This prospective observational study evaluated changes in the immune system in response to treatment by measuring gene expression levels in PBMCs before and after eight sessions of controlled manual therapy (MT) (two weekly, as detailed in Methods) on FM patients (n = 38) (NCT04174300 clinical trial) [18] and non-FM volunteers (n = 12). To assess participant baseline status and monitor response to therapy, questionnaire scores were obtained with the fibromyalgia impact questionnaire (FIQ) [19,20], multi-fatigue inventory (MFI) [21], and the SF-36 quality of life instrument (Likert scale) [22]. In addition, pressure point thresholds (PPTs) at the baseline and after treatment were measured in FM, as previously described [18], as well as in the non-FM “control” cohort.

#### 2.1.1. Demographics of Participants by Study Cohort

The patient cohort included 38 FM patients (thirty-five females and three males) who fulfilled 1990 and/or 2010 ACR criteria [2,3], 50% (19/38) of them presenting comorbid ME/CFS according to the Canadian and/or international diagnostic criteria [5,6], as previously described [18]. Due to the frequent symptom overlap with post-COVID-19 conditions (popularly known as long COVID) [23], it should be highlighted that the FM cohort studied is pre-pandemic (NCT04174300 completed before 03/2020). The average age for the FM cohort was 55.6 ± 7.2 years (range 43–71), and the time from primary FM diagnosis over 3 years was 10.3 ± 7.5 years (range 3–21). A subcohort of six participants composed entirely of women, with an average age of 54 years ± 8.44 and a range of between 43 and 69 years, was selected for PBMC RNAseq analysis to evaluate the effects of the MT program on the immune system of FM (see Section 2.2 for details).

On another side, a non-FM matched cohort of 12 female participants with an average age of 52.33 ± 6.2 years (range 43–61) was subjected to the same MT protocol to investigate whether gene expression changes were specific for FM or, by contrast, corresponded with a generic response to MT. In this “control” non-FM cohort, only three participants had a diagnosed pathology, assuming 25% of the sample. Pathologies were diabetes in one participant (8.33%), and osteoarthritis in two participants (16.66%), and none had ever received an FM or ME/CFS diagnosis.

#### 2.1.2. Participant Phenotyping

As mentioned in the study design section above, all participants were finely phenotyped with the use of the FIQ [19,20], MFI [21], and SF-36 (Likert scale) [22] validated questionnaires. As shown in Table 1, participating patients presented severe FM (total FIQ > 59) and moderate fatigue, with most domains showing scores well above the non-FM control cohort, except for general fatigue, where scores were very similar. Quality of life (SF-36) was much superior (>50 in all domains) in the non-FM cohort than the FM cohort, except for the role emotional domain. No major baseline (pretreatment) differences were found between the subcohort of FM (n = 6) subjected to RNAseq analysis and the previously described complete FM cohort (n = 38) [18], while the non-FM cohort statistically differed from the complete, as well as the sequenced subcohorts of FM patients. Individual participant scores are available in Appendix A.

In addition, a comparison of baseline PPTs showed that low cervical points were the most sensitive to pressure-induced pain, while gluteus, trochanters, and knees were the least sensitive to pain in both FM cohorts, with a tendency to increased sensitivity in the trapezius right point for the RNAseq FM subcohort (n = 6) (*p*-value = 0.052, Table 2). As expected, non-FM participants showed marked differences in resistance to pressure-triggered pain (Table 2). Individual participant PPT values are available in Appendix A.

### 2.2. Differential Gene Expression in PBMCs of FM with Therapy

Differential gene expression in PBMCs of FM with MT was assessed by RNAseq analysis of total RNA prepared from an RNAseq subcohort of FM patients (discovery phase) before and after the MT treatment (n = 12 paired samples from six patients). The results showed an overexpression of 72 transcripts and an underexpression of 256 transcripts (*p* < 0.05, FDR < 0.1) (Figure 1, and Appendix A).

At the individual level, significant changes with treatment varied across participants, with the upregulation of at least 22 genes and the downregulation of at least 42 in all participants (see Appendix A for individual volcano plots and Appendix A for RNAseq DE analysis at the individual level).

### 2.3. Gene Enrichment and Pathway Analysis with MT in the Immune System of FM

To understand the biological significance of the genes differentially expressed (DE) with MT in PBMCs of FM, we performed enrichment analysis using the gene ontology (GO) knowledgebase (http://geneontology.org, accessed on 31 July 2024) [24,25] (Figure 2, left panel) and the Kyoto Encyclopedia of Genes and Genomes (KEGG) database (https://www.genome.jp/kegg/pathway.html, accessed on 31 July 2024) (Figure 2, right panel), as detailed in Methods.

The findings included responses to stress and infectious processes, with the involvement of cytokine, chemokine, MAPK, and NFkB signaling processes (Figure 2 and Appendix A).

### 2.4. RT-qPCR Validation of Protein-Coding Genes Differentially Expressed in Response to MT in FM

To validate top relevant functions among the 328 differentially expressed (DE) transcripts with treatment (*p* < 0.05, FDR < 0.1) that are now in the complete FM cohort of (n = 38), we selected only RNAs with protein-coding potential (Appendix A, coding potential tab) by removing 149 TCONS (Transcript Cluster Noncoding RNAs), 11 pseudogenes, one miRNA cluster, two lncRNAs (RP4-737E23.5 and RP11-213H15.1), three lincRNAs (*LINC00861*, *LINC00877*, and *AP001046.5*), and seven novel transcripts. Therefore, RNAs with coding potential DE by MT included 161 DE RNAs (22 upregulated and 139 downregulated). Then, we evaluated DE at the individual level and imposed the condition of having DE in at least 50% of the samples (n ≥ 3), (Table 3), which reduced the list to only six genes being consistently overexpressed and eighteen underexpressed by MT. They were then considered top candidate effectors of the therapy. Individual DE data are provided in Appendix A and summarized in Table 3 (Indiv.pval < 0.05, FDR < 0.1).

GO and KEGG analyses were then reassessed for these top 24 DE RNAseq genes of individual responses to MT in at least 50% of the FM cases, finding that responses to bacteria and chemokine signaling were among top cell functions affected by MT (Figure 3).

To orthogonally validate DE RNAseq genes by the alternative RT-qPCR method, primer sets were designed with even more restrictive selection criteria, as the amount of material for this step was limited. To select the top six DE genes out of the twenty-four listed in Table 3, we chose the least individually altered with MT in at least four out of the six FM cases from the RNAseq subcohort. With this, we designed primers for *CD3E* and *CX3CR1* from the upregulated group. However, for the downregulated group, the number of genes fulfilling the applied criteria (being DE in at least 4 out of the 6 patients) exceeded four genes (Table 3, downregulated), among those the top 4 candidates were *SIK1*, *HBEGF*, and *EGR2*, all individually downregulated in five of the six FM cases. In addition, we also included the top downregulated gene (*EREG*) in this RT-qPCR validation step (Table 3). Individual details for DE of these genes are provided in Appendix A. Primer sets to measure the expression levels for these six DE-selected genes (Appendix A), together with a set to detect the housekeeping *GAPDH* gene, were then used for technical validation of the RNAseq results in our FM subcohort (n = 6), as well as in the complete FM cohort (n = 38) (population validation of our RNAseq data).

The upper panel of Figure 4 illustrates individual relative DE values of the protein-coding genes selected for RT-qPCR validation, according to RNAseq individual data (Appendix A) and the stringent selection criteria described above, while the lower panel in Figure 3 shows the results obtained by the alternative RT-qPCR approach, performed to validate the RNAseq results in the FM subcohort (n = 6). As observed, only *SIK1* and *HBEGF* downregulation could be validated. None of the four remaining selected DE genes appeared significantly changed with MT by RT-qPCR (Figure 3, lower panel).

However, RT-qPCR analysis of the complete FM cohort (n = 38) confirmed the upregulation of *CX3CR1* and the downregulation of all four selected genes among those downregulated by RNAseq. No significant change with MT was found for *CD3E* by RT-qPCR (Figure 5, upper panel).

To investigate whether DE with MT were exclusive to FM or their change with MT corresponded to a generalized response, DE was also measured in PBMCs from non-FM matched volunteers subjected to the same eight-session MT treatment program previously described [18] (n = 12). As can be observed in the lower panel of Figure 5, while the upregulation of *CX3CR1* seemed also upregulated and *HBEGF* appeared downregulated by MT (interpreted as potential general responders to MT), all the three other genes downregulated in FM by MT (*EREG*, *EGR2*, and *SIK1*) did not show significant changes in non-FM individuals, interestingly, suggesting that these genes may constitute sensors of the response to MT therapy in FM. It should be noted that basal expression levels vary greatly across the two groups being compared, with an approximately 10-fold difference for *CX3CR1* and *HBEGF*, which increased in the FM group and decreased in the indicated group for *EREG*. The potential significance of these large basal differences is unknown. However, *SIK1* basal levels appear similar between the FM and the non-FM cohorts but are only significantly inhibited by MT in the FM group.

Furthermore, since our previous work, in the context of clinical trial NCT04174300, identified differences in response to MT among FM patients co-diagnosed with Myalgic Encephalomyelitis/Chronic Fatigue Syndrome (ME/CFS) [18], we reassessed our RT-qPCR results, taking into account whether or not patients with FM had also received the ME/CFS diagnostic (Appendix A). The results indeed point out that the FM group with ME/CFS co-diagnosis (n = 19) does not seem to respond to MT by increasing their *CX3CR1* levels, while DE of *HBEGF* and *EGR2* appears more related to this group (Figure 6). *EREG* and *SIK1* DE seem to specifically associate with both patient groups, without changes in the control non-FM group (Figure 5).

### 2.5. Correlation of Genes Differentially Expressed in Response to MT with Patient Symptoms and Sensitivity to Pain (PPTs)

MT seems to provide improvement of patient symptoms to a certain extent, as shown by the changes detected for overall and total FIQ scores, as well as for the SF-36 “Bodily pain” domain for the complete FM group (n = 38), as previously reported [18]. By contrast, changes were not appreciated in the FM RNAseq RNAseq cohort (n = 6), perhaps due to the small sample size, and were absent in the non-FM cohort (n = 12), as expected (Table 4). In the latter, scores seem to rather be associated with improved fatigue and mental health, perhaps related to the relief of some ailments unrelated to FM.

On another side, the monitoring of PPT changes with MT also had shown significant changes with treatment for the most sensitive “Low cervical” tender points (n = 38) [18]. Again, no differences were detected for the FM subcohort (n = 6), and unexpectedly, thresholds changed with treatments for the non-FM cohort in different anatomic locations (Table 5).

To determine if the DE genes with MT play a role in improvement of symptoms, we evaluated the potential correlations between gene expression and symptom differences across our validated results. Figure 7, interestingly, shows that low levels of *SIK1* negatively correlate with higher scores for the SF-36 “Bodily pain” domain, which indicates better health (reduced pain), supporting a potential role of the MT treatment in reducing pain in FM by decreasing SIK1 levels. Positive correlations of *SIK1* levels with FIQ “Overall” and “Symptoms” domains also support their potential participation in treatment-associated patient improvement. By contrast, high MFI “Physical fatigue” domain scores associated with lower *SIK1* levels would indicate worsening of fatigue in FM with treatment (Figure 7, upper left panel). Importantly, no relevant correlations of *SIK1* levels with questionnaire score changes after treatment were detected in the control group (Figure 7, upper right panel), supporting a specific role of SIK1 in FM symptom relief in FM with MT.

Although the correlations with total FIQ and SF-36 vitality with the upregulation of *CX3CR1* and those of *HBGEF* with the same domains correlating with *SIK1*, except for the SF-36, seem to mainly associate with symptoms improvement with a change in expression, the observed changes in these two genes are not specific to FM (Figure 5).

On another end, lower levels of *SIK1* seem to correlate with increased threshold values in the lower cervical left tender point of FM patients (n = 38) (Figure 7, lower left panel), indicating that the changes in SIK1 may be associated with improvement of patient allodynia. No correlations of the changes in *SIK1* with PPT value improvement were detected in the control non-FM group (n = 12) (Figure 7, lower right panel). Statistically significant correlations of PPT changes in other DE genes (*CX3CR1* with low cervical right and *EREG* with gluteal left) were considered spurious since DE of *CX3CR1* and *EREG* were found to be non-significant by RT-qPCR in this control group (Figure 5), and, therefore, their detailed analysis was not further pursued.

As differences in response to MT in patients having received ME/CFS co-diagnosis had been previously reported [18], we set to evaluate potential correlations between DE genes and symptoms or between DE genes and PPT changes with ME/CFS co-diagnosis. As shown in Figure 8 (upper left), patients not fulfilling diagnosis criteria for ME/CFS show a clear benefit of the MT program applied, mostly reproducing the results obtained with the full cohort. By contrast, the subgroup of FM participants that had received a diagnosis of ME/CFS as well seemed to present a more reduced benefit of symptoms associated with decreased levels of *SIK1* (Figure 8, upper right), with significant improvement of the symptom domain of the FIQ questionnaire for low *SIK1* levels only. Again, significant associations between symptoms and other DE genes, different from *SIK1*, were no further pursued, as DE of the gene had not passed the test of significance (e.g., *CX3CR1* in the ME/CFS co-diagnosed group or *EGR2* in the FM subgroup not having received co-diagnosis of FM) (see Figure 6) or had shown lower correlation values (e.g., *CX3CR1* with mental health of the SF-36 questionnaire) (Figure 8, upper left).

With respect to changes in PPT values with reduced *SIK1* levels, once more, the FM subgroup not fulfilling ME/CFS diagnosis criteria seems to behave as the complete FM group, indicating that 50% of patients in the group (FM with ME/CFS diagnosis) were mostly irresponsive to MT, at least symptom wise with *SIK1* changes. Statistically significant associations unrelated to *SIK1* were not further examined, as the affected PPTs (e.g., great trochanters or lateral epicondyle humerus) were not among the affected by MT in FM (Table 5).

## 3. Discussion

This study expands our previous knowledge on the improvement of FM symptoms by a self-designed controlled-pressure MT protocol (NCT04174300) [18] by evidencing molecular changes in the immune system of FM participants with MT. This study comprised two phases: a discovery phase of genome-wide transcriptomic profiling to detect changes in expression levels with MT in the immune system of an RNAseq FM subcohort (n = 6) and a validation phase, extending the main molecular findings to the complete cohort (n = 38). This later validation phase also examined changes in expression levels with MT in the immune system of a non-FM control cohort (n = 12) treated with the same self-designed controlled-pressure MT protocol as the FM cases [18]. The objective was to find out if the observed findings in the immune system with MT were specific to FM or, by contrast, corresponded with a general mechanism triggered by MT in all individuals. Although limitations associated with the selection process of an RNAseq subcohort of FM and the selection of DE genes with MT leave room for further findings, the results strikingly show that the downregulation of *SIK1* correlates with patient symptom improvement, particularly with some FIQ domains (“Symptoms” and “Overall”), as well as with the SF-36 “Bodily pain” subdomain, with the latter two domains having shown most improvement with MT [18]. They also show that this correlation is specific for FM, as *SIK1* levels do not seem to change with MT in the immune system of control non-FM participants. Whether and how ***SIK1*** changes affect FM immune transcriptome [16,17] will require further work. However, our finding that basal levels between the FM and non-FM cohorts appear similar in our RT-qPCR data and the absence of a direct connection of *SIK1* with FM phenotype, to the best of our knowledge, may indicate an indirect effect of SIK1 inhibition to mediate MT therapeutic action.

SIK1, initially identified for its role in sodium sensing, belongs to the salt-inducible kinases (SIKs) family, which includes three homologous serine–threonine kinases (SIK1, SIK2, and SIK3) that regulate multiple aspects of the human physiology in response to extracellular signals, including feeding/fasting metabolic responses, inflammation and immune responses, and sleep (circadian rhythms), among others [26]. Of the three kinases, only *SIK1* expression is upregulated at the transcriptional level through a consensus CREB (cAMP response element) present in its promoter, as shown in myocytes and the suprachiasmatic nucleus of the brain (SCN) [26,27,28]. SIK activity regulates innate immunity responses by suppressing the production of the IL-10 anti-inflammatory cytokine in macrophages. In fact, pharmacological inhibition of SIK activity increases the levels of IL-10 while suppressing the levels of the proinflammatory IL-6, IL-12, and TNF-α after TLR (toll-like receptor) stimulation by LPS [27,29]. However, conflictive data regarding the production of proinflammatory cytokines and activation of the transcription factor NFκB with increased SIK activity exist [30,31].

The current intense research in the development of member-specific inhibitors of SIK activity [32,33,34] should eventually help to ascertain the precise attributes and contributions of each member of this family of proteins, in particular cell and environmental scenarios, leading to the development of novel pharmacological treatments. For example, to increase the production of IL-10 in the gut, Sundberg et al. screened a library of kinase inhibitors after challenging murine bone-marrow-derived dendritic cells (DCs) with the yeast cell wall preparation zymosan, finding that the protective effects involved SIK activity inhibition in a subpopulation of CD11c (+) CX3CR1(hi) cells isolated from murine gut tissue [32]. Thus, SIK activity seems relevant in still other immune system compartments, including mast cell IL-33 cytokine release [35], and it even modulates adaptive immunity through the regulation of T-cell lineage commitment, differentiation, and survival [36,37], therefore offering a multitude of potential indirect mechanisms to exert MT therapeutic effects in FM. Although drastic SIK1 downregulation may not be desirable because of its role in blood pressure or its tunning in certain cell types or diseases could constitute valued therapeutic options [30,38,39].

Whether *SIK1* transcription downregulation by MT is mediated through the conserved CREB element in its promoter or through alternative mechanisms seems an important question for future work in the field of physiotherapy. Other possibilities worth exploring after this initial finding are the potential impact of MT on the muscle, blood pressure, and cell metabolism or on the circadian system through changes in SIK activity.

By contrast, *CX3CR1* levels appeared significantly increased in both study groups (FM and non-FM individuals), indicating that MT triggers this change in all individuals, with the exception of those FM patients co-diagnosed with ME/CFS. CX3CR1 is a G-protein-coupled receptor and the only binder of fractalkine present on a subset of immune cells, including monocytes and macrophages, as well as DCs, T helper (Th) 1, CD8+T effector/memory and γδ T lymphocytes, and NK cells [40]. Its main role in immune cells is to detect and migrate toward inflamed tissue, “crawling and patrolling” from blood vessel endothelium to different destinies according to fractalkine’s gradient, the objective being to initiate innate immune responses followed by adaptive responses [40,41]. In the brain, it is mainly expressed in astrocytes and microglia regulating cellular communication between neurons, in addition to providing protection from the neurotoxicity induced by the HIV-1 envelope protein gp120 [42]. In the gut, CX3CR1-positive macrophages produce the IL-10 immunoregulatory cytokine [43], and a lack of CX3CR1 expression is associated with an altered microbiome and impaired intestinal barrier [44]. Regulatory mechanisms of CX3CR1 expression and the implications of its overexpression are complex and require further research to understand their impact on health and disease. Why patients co-diagnosed with ME/CFS do not respond to MT with increased *CX3CR1* levels is unknown at present, but it seems to support differential response to MT with ME/CFS co-diagnosis, as previously shown [18].

Molecular basal differences between patients fulfilling only FM or ME/CFS (co-diagnosis) have been found by our group [45] and by others [46], seemingly demanding a review of the case definition for patients fulfilling both clinical criteria [45]. Our previous report, NCT04174300 [18] showed differences in response to MT between patients that had or had not received a co-diagnosis of ME/CFS. The results of this study further confirm differences across these two FM subgroups, not only for a lack of upregulation of *CX3CR1* levels in response to MT but also for the downregulation of *EGR2*, occurring only in the co-diagnosed group. EGR2 or early growth response 2 is a transcription factor with an essential epigenetic regulatory role (DNA methylation turnover) for the differentiation of human monocytes [47] and a novel regulator of the senescence of fibroblast and epithelial cells [48]. Together with EGR3, it is needed for T- and B-cell development and activation [49]. Whether MT preferential upregulation of EGR2 in patients with an ME/CFS status relates to increased *EGR2* basal levels in these patients (as shown in Figure 6), coinciding with Dr. Kerr’s previous findings [50,51], and whether this relates to EBV infection history of the patient, seems like a possibility to be further explored.

Finally, the downregulation of *EREG*, also known as epiregulin, seems to discriminate responses to MT between both FM subgroups and the control. Being a soluble peptide hormone involved in inflammation and wound healing, upregulated by LPS induction and by stress of the endoplasmic reticulum [52], its downregulation by MT may relate to patient improvement. However, correlations with questionnaire scores did not detect such a link.

The fact that RT-qPCR did not validate the increased levels of *CD3E* with MT detected by RNAseq does not serve to refute its findings, as the methodological differences may indeed constitute the reason for the discrepancy.

On the question of what could the mechanisms that exert changes in the molecular profiles of immune cells by MT be, we are far from being able to give a detailed response. However, elucidation of the immunomodulatory effects of massage either by direct pressure/mechanotransduction or by indirect pathways effected by MT, such as cytokine, chemokine, microRNA release [12], sleep improvement [53], or others, are on their way.

## 4. Materials and Methods

### 4.1. Study Design and Intervention

This is an observational study consisting of the analysis of the molecular changes taking place in the circulating immune cells of FM patients (n = 38) and non-FM patients (n = 12) by a physiotherapy program of manual therapy. The program consisted of eight sessions (twice weekly for four weeks) with a 25 min custom protocol, including pressure maneuvers of about 4.5 N each of 10 out of the 18 FM tender points and surrounding areas (NCT04174300), as described in our previous publication [18]. This study also included a pilot interventional non-randomized single-arm trial, replicating the treatment program previously applied to FM and now using a matched non-FM cohort as a control. Participants could not be enrolled in pharmacological CTs or receive additional physiotherapy treatment while participating in this study and agreed to withdraw medication 12 h before blood draws. Comparison across groups was performed to find out whether the MT program triggers similar or distinct changes in FM vs. non-FM individuals. Before vs. after FIQ [19,20], MFI [21], and SF-36 [22] questionnaire scores, as well as PPT values of the 18-FM tender points, were registered for all participants. The studies were approved by the Universidad Católica de Valencia San Vicente Mártir Ethics Committee with study codes UCV/2018-2019/076 and UCV/2020-2021/167, respectively. All participants signed an informed consent before they were included in this study. For the analysis of molecular changes in the immune system of FM, the whole transcriptome of an RNAseq subcohort of FM patients (n = 6) was obtained before and after the program. Top differentially expressed genes were validated in the complete FM cohort (n = 38) by RT-qPCR and studied in the non-FM cohort (n = 12) for their comparison.

### 4.2. Total RNA Preparation and Quality Assessment

Total RNA was prepared from a previous collection of PBMC pellets (−150 °C) (≥10^6^ cells) registered at the Institute of Health Carlos III National Biobank, Madrid, Spain (Ref. C.0006924) [12] with the RNeasy Mini Kit (Qiagen, Venlo, The Netherlands cat. 74104) following the manufacturer’s protocol. Cell lysis included the addition of 1/100 β-mercaptoethanol (Sigma, Burlington, MA, USA cat. 63689) before a 5 min vortex to favor lysis. Removal of contaminant DNA was performed in a column with the RNase-free DNase set (Qiagen, cat. 79254). Elution RNase-free ddH_2_O was supplemented with RNasin (Promega, Madison, WI, USA cat. N2118) to a final concentration of 0.4 U/µL before use. RNA yields and quality were obtained with an Agilent TapeStation 2100 (Agilent Technologies, Santa Clara, CA, USA). RNA integrity was double checked by agarose gel electrophoresis and individual electropherograms (Appendix A). Only samples with RIN ≥ 7 were subjected to downstream analysis.

### 4.3. RNAseq

Upon ribosomal RNA depletion, 1 µg of RNA was used for whole transcriptome sequencing (Illumina NovaSeq 6000 PE150, 50M reads) (Novogene, Cambridge, UK). After the removal of filtered to adapter sequences and low-quality reads, sequences were aligned to the human GRCh38/hg38 genome using HISAT2 software v.2.2.0 [54]. Sequence assembly analysis was performed using Cufflinks, converted to BAM format, and sorted and indexed with samtools [55]. Cuffdiff was used to calculate differential expression, expressed as FPKM (fragments per kilobase of exon per million mapped fragments) numbers for each sample (https://cole-trapnell-lab.github.io/cufflinks/cuffdiff/index.html, accessed on 30 March 2020). Differences were given as the absolute value log2 fold change in the ratio between the pretreated and the posttreatment samples > 1, *p*-adjust < 0.05, and FDR < 0.1 = TRUE. Heatmaps and volcano plots were generated with the CummeRbund R package (R version 4.2.1, cummeRbund 2.38.0) [56] and GraphPad Prism 8.0.2 software.

### 4.4. Enrichment Analysis

Gene ontology (GO) enrichment analysis of differentially expressed genes was performed with the gene ontology (GO) knowledgebase (http://geneontology.org, accessed on 31 July 2024) [24,25] or with the Goseq, version 1.56.0 and R package, version 4.4 [57]. After gene length bias correction, *p*-values less than 0.05 were considered significantly enriched by differentially expressed genes.

For pathway analysis of differentially expressed genes, the Kyoto Encyclopedia of Genes and Genomes (KEGG) analysis of differentially expressed genes was performed with the https://www.genome.jp/kegg, accessed on 31 July 2024, online tool.

### 4.5. RT-qPCR Validation

Reverse transcription was performed using the High Capacity Complementary DNA (cDNA) Reverse Transcription Kit (Applied Biosystems, Waltham, MA, USA, cat. 4308228) with 1–2 μg of total RNA, according to the manufacturer’s guidelines. Relative gene expression was assessed by qPCR of triplicates per sample, with the primer sets in Appendix A. qPCR was performed with the PowerUP Sybr Green Master Mix (Applied Biosystems, Foster City, CA, USA cat. 100029283) on a Lightcycler LC480 instrument (Roche, Penzberg, Germany) with the following amplification conditions: a single preactivation cycle of the hotstart polymerase at 94 °C for 15 min, followed by 40 amplification cycles, each of which consisted of three steps: 95 °C for 15 s, 54 °C for 30 s, and extension at 70 °C for 30 s. Gene expression levels were normalized to GAPDH and quantified by the 2^−∆∆Ct^ method [58].

### 4.6. Statistics

Continuous data are expressed as means ± SD (standard deviation) and range values. Statistical differences were assessed by paired *t*-tests for normal value distributions and either nonparametric Mann–Whitney or Wilcoxon analysis if values did not follow a normal distribution. Normality was determined with the Shapiro–Wilk test. Differences between groups were considered significant when *p* ≤ 0.05. Analysis was conducted with Excel, the SPSS package 13.0 (SPSS Inc., Chicago, IL, USA), and R v4.2.1 [59]. Pearson correlations were evaluated with the WGCNA R package v1.72.5 [60]. Plots were drawn using the GraphPad Prism 5.0 program (San Diego, CA, USA) and the package ggplot2 [61].

## 5. Conclusions

In conclusion, the molecular data obtained by comparing the immune system transcriptome of FM before and after a pressure-controlled MT protocol (NCT04174300) identifies the downregulation of *SIK1* as a prominent and specific target of MT in FM, which is associated with patient symptom improvement. In addition to SIK1’s potential biomarker value for monitoring the response to MT in FM, pending external validation in extended cohorts, this pioneer finding opens the interesting possibility of therapeutic applications of SIK1 inhibitors on FM. Future research efforts in the direction of improving FM healthcare seem granted.

## Figures and Tables

**Figure 1 ijms-25-09523-f001:**
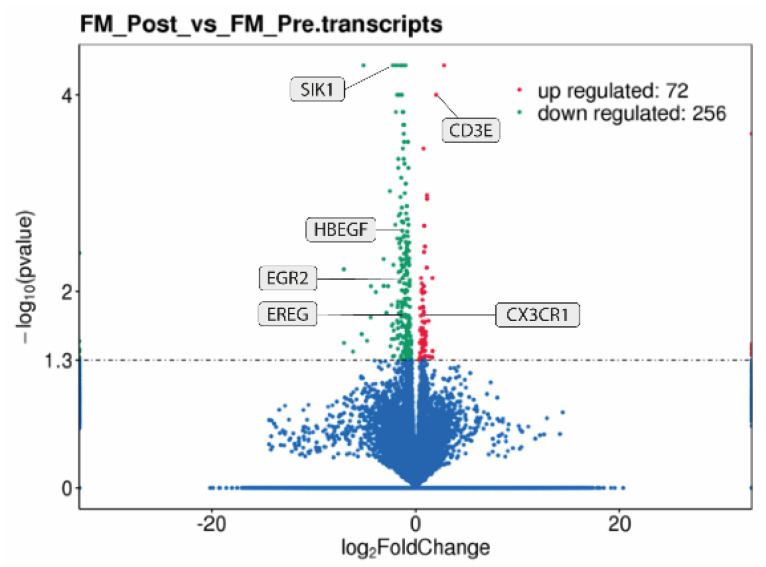
Volcano plot representation of differential gene expression in PBMCs of FM with therapy. |Log2FoldChange| > 1 values (*X* axis) are displayed with respect to log10 of their *p*-values (*Y* axis), and the significance is set at *p* < 0.05, FDR < 0.1.

**Figure 2 ijms-25-09523-f002:**
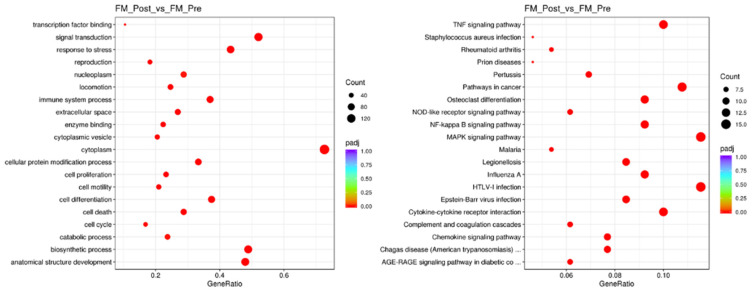
GO (**left**) and KEGG (**right**) pathways targeted by MT in the immune system of FM. Function significance (color palette, padj < 0.05), DE gene count in each pathway are indicated by dot thickness for each panel.

**Figure 3 ijms-25-09523-f003:**
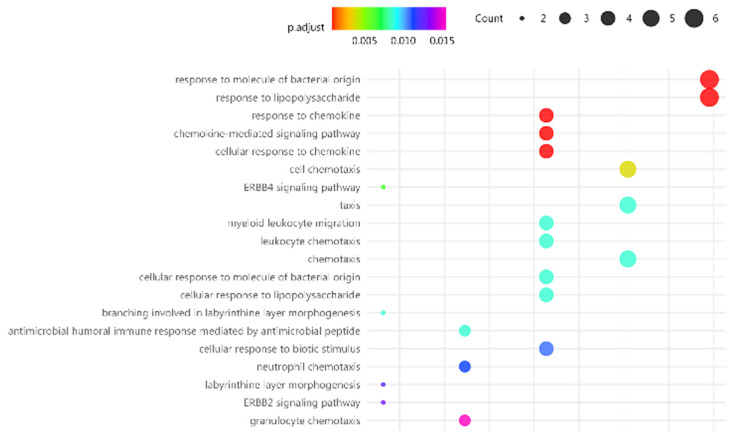
Top pathways targeted by MT in the immune system of FM, as determined by top RNAseq DE genes. Function significance (color palette, padj < 0.05) and approximate DE gene count in each pathway are indicated by dot thickness for each panel.

**Figure 4 ijms-25-09523-f004:**
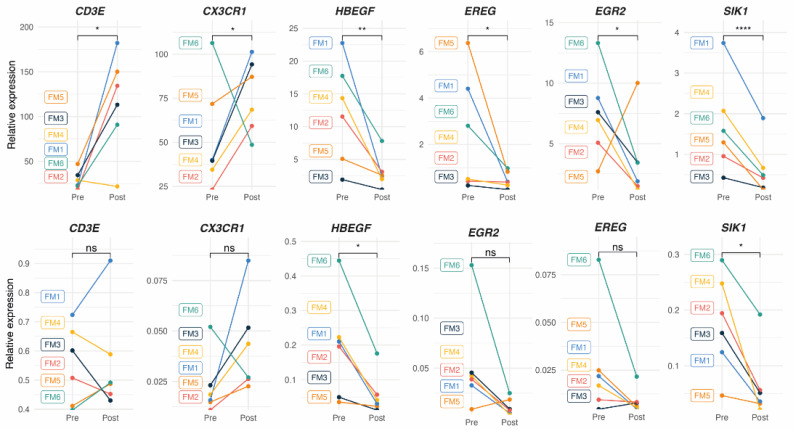
Relative expression levels (pre vs. post) of randomly selected DE coding genes with MT in the RNAseq subcohort of FM patients at the individual level, as determined by RNAseq analysis (**upper** panel) (Appendix A, *p* < 0.05; FDR < 0.1), RT-qPCR analysis (**lower** panel) (n = 6), and a Wilcoxon test (ns, non-significant, * *p* < 0.05, ** *p* < 0.01, **** *p* < 0.0001).

**Figure 5 ijms-25-09523-f005:**
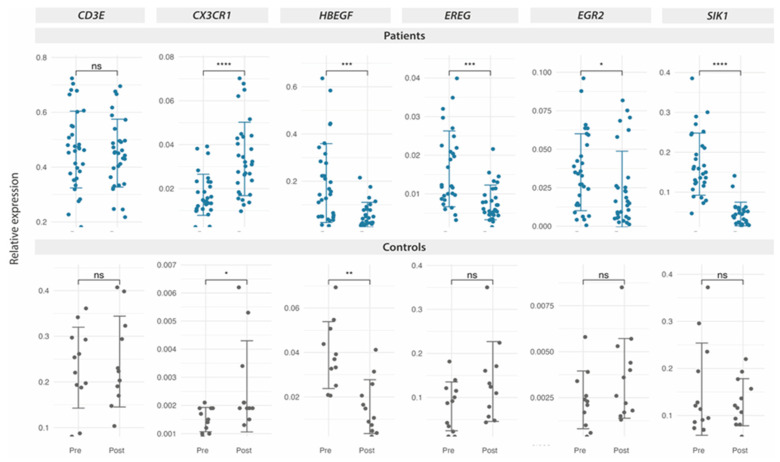
DE-expressed genes with MT on PBMCs from FM patients, as determined by RT-qPCR. Relative expression by ΔΔCt values upon *GAPDH* normalization for each sample (triplicates) in each study group (n = 38 for the FM cohort in blue, upper panel, and n = 12 for the non-FM control cohort in black, lower panel) are shown. A statistical paired two-Wilcoxon test with Benjamin-Hochberg *p*-value correction. (ns, non-significant, * *p* < 0.05, ** *p* < 0.01, *** *p* < 0.001, **** *p* < 0.0001) was applied to assess the significance of DE.

**Figure 6 ijms-25-09523-f006:**
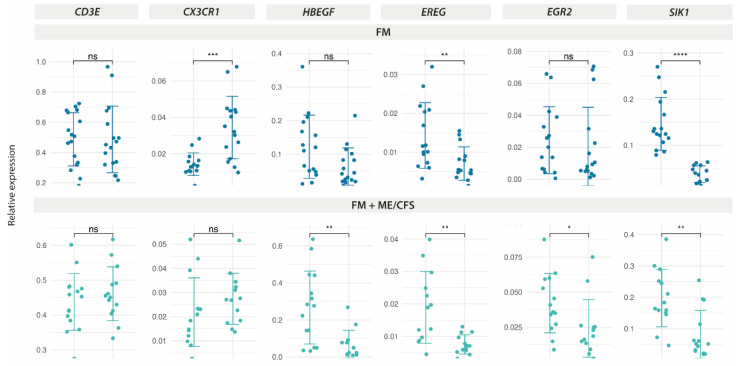
DE-expressed genes with MT on PBMCs from FM patients with or without ME/CFS co-diagnosis, as determined by RT-qPCR. Relative expression by ΔΔCt values upon *GAPDH* normalization for each sample (triplicates) in each study group (n = 19 for the FM-only group, dark blue, **upper** panel; and n = 19 for the FM group with ME/CFS co-diagnosis, light blue, **lower** panel) are shown. A statistical paired two-Wilcoxon test with Benjamin–Hochberg *p*-value correction. (ns, non-significant, * *p* < 0.05, ** *p* < 0.01, *** *p* < 0.001, **** *p* < 0.0001) was applied to assess the significance of DE.

**Figure 7 ijms-25-09523-f007:**
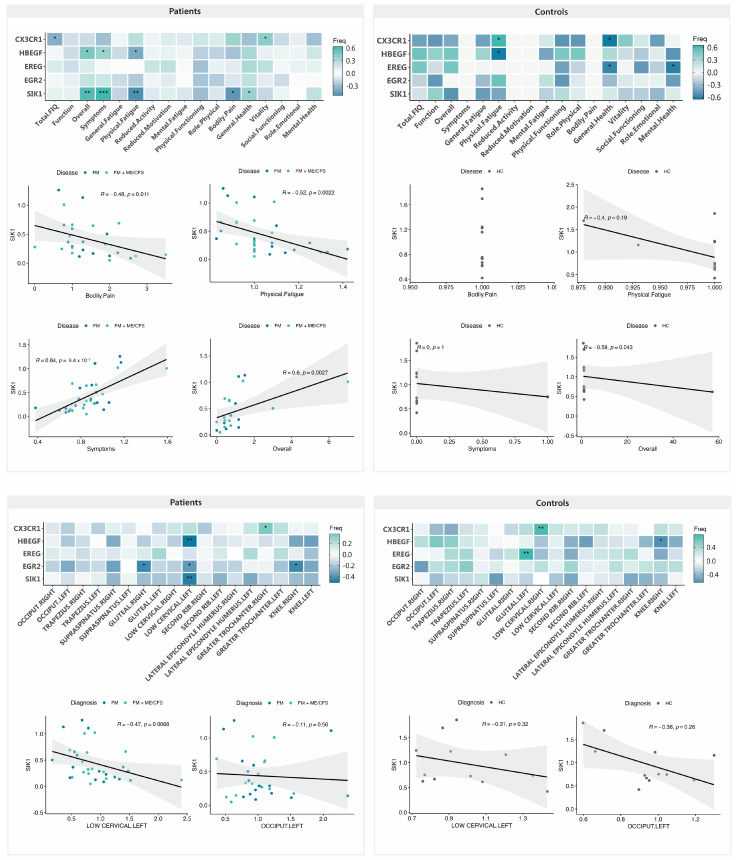
Symptom improvement with validated DE genes with MT in FM (n = 38) (**upper left**) and non-FM controls (n = 12) (**upper right**), and the correlation of PPT ratios (post and pre) with DE genes in FM (n = 38) (**lower left**) and non-FM controls (n = 12) (**lower right**). Pearson correlation values and associated *p*-values (*, *p* < 0.05; **, *p* < 0.01; ***, *p* < 0.001) between gene expression levels and symptom scores or PPT ratios are shown.

**Figure 8 ijms-25-09523-f008:**
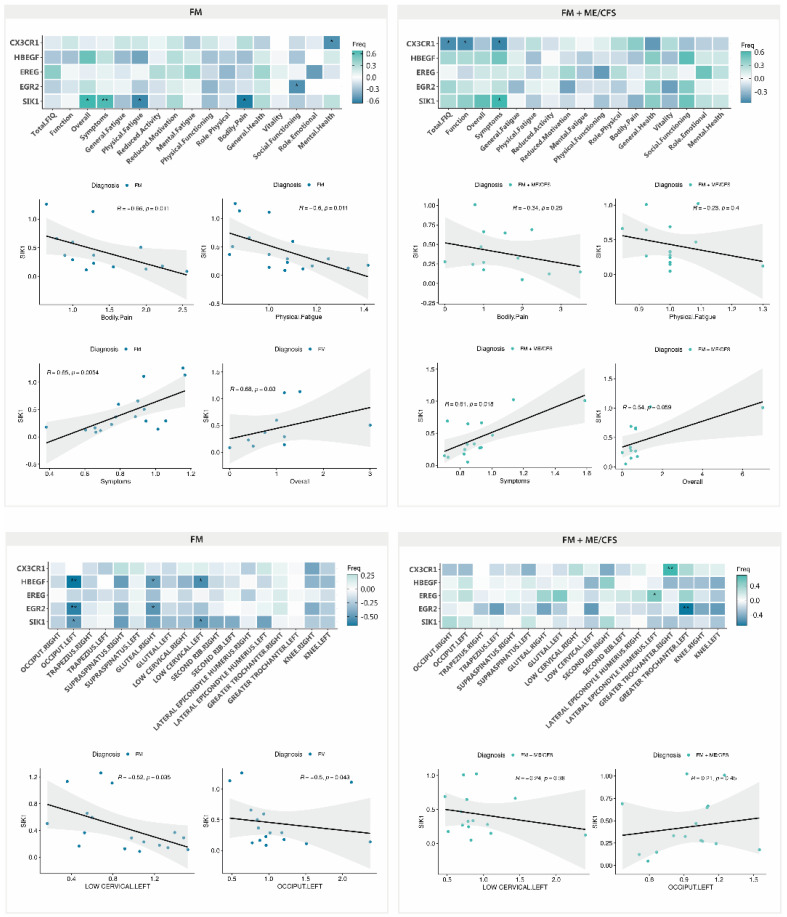
Symptom improvement with validated DE genes with MT in FM (n = 19) (**upper left**) and FM with co-diagnosis of ME/CFS (n = 19) (**upper right**), and the correlation of PPT ratios (post and pre) with DE genes in FM (n = 19) (**lower left**) and FM with the co-diagnosis of ME/CFS (n = 19) (**lower right**). Pearson correlation values and associated *p*-values (*, *p* < 0.05; **, *p* < 0.01) between gene expression levels and symptom scores or PPT ratios are shown.

**Table 1 ijms-25-09523-t001:** Participant baseline FIQ [19,20], MFI [21], and SF-36 (Likert scale) [22] questionnaire scores by cohort, as indicated.

Questionnaire	Total Cohort (*n* = 38) Mean Pre- ± SD [Range]	RNAseq RNAseq Cohort (*n* = 6) Mean Pre- ± SD [Range]	Non-FM Cohort (*n* = 12) Mean Pre- ± SD [Range]	*p*-Value (1)	*p*-Value (2)	*p*-Value (3)
**FIQ**						
Total FIQ	72.62 ± 15.67 [41.08–96.51]	80.05 ± 17.26 [46.12–92.09]	27.29 ± 13.28 [5.64–49.64]	0.269	**<0.001**	**0.005**
Function	5.16 ± 2.29 [0–9.24]	5.72 ± 2.11 [3.3–8.25]	2.39 ± 0.50 [1.98–3.63]	0.492	**0.016**	**0.012**
Overall	8.30 ± 2.23 [2.86–10.01]	8.10 ± 1.73 [5.72–10.01]	9.17 ± 2.88 [0–10.01]	0.817	0.530	**0.043**
Symptoms	4.59 ± 3.72 [0–10.01]	7.39 ± 3.77 [0–10.01]	0.47 ± 1.11 [0–2.86]	0.079	**0.014**	**0.005**
**MFI**						
General Fatigue	11.5 ± 1.6 [7, 8, 9, 10, 11, 12, 13, 14, 15, 16]	11 ± 1.79 [8, 9, 10, 11, 12, 13]	11.41 ± 4.21 [5, 6, 7, 8, 9, 10, 11, 12, 13, 14, 15, 16, 17]	0.274	0.694	0.211
Physical Fatigue	12.3 ± 1.2 [10, 11, 12, 13, 14, 15, 16]	12.83 ± 1.17 [12, 13, 14, 15]	9.91 ± 3.75 [6, 7, 8, 9, 10, 11, 12, 13, 14, 15, 16, 17]	0.741	**0.023**	0.218
Reduced Activity	12.1 ± 1.9 [6, 7, 8, 9, 10, 11, 12, 13, 14, 15, 16, 17, 18, 19]	11.5 ± 2.74 [6, 7, 8, 9, 10, 11, 12, 13]	8.08 ± 2.87 [4, 5, 6, 7, 8, 9, 10, 11, 12, 13, 14]	1.000	**0.001**	**0.013**
Reduced Motivation	10.6 ± 2.7 [4, 5, 6, 7, 8, 9, 10, 11, 12, 13, 14, 15, 16, 17, 18, 19]	10.83 ± 4.54 [6, 7, 8, 9, 10, 11, 12, 13, 14, 15, 16, 17, 18, 19]	7.58 ± 3.14 [4, 5, 6, 7, 8, 9, 10, 11, 12, 13, 14, 15]	0.920	0.082	**0.009**
Mental Fatigue	11.5 ± 1.8 [7, 8, 9, 10, 11, 12, 13, 14, 15]	11.5 ± 1.52 [10, 11, 12, 13, 14]	7.08 ± 3.34 [4, 5, 6, 7, 8, 9, 10, 11, 12, 13]	0.363	**<0.01**	**0.033**
**SF-36**						
Physical Functioning (PF)	38.95 ± 17.48 [0–85]	37.5 ± 16.96 [15, 16, 17, 18, 19, 20, 21, 22, 23, 24, 25, 26, 27, 28, 29, 30, 31, 32, 33, 34, 35, 36, 37, 38, 39, 40, 41, 42, 43, 44, 45, 46, 47, 48, 49, 50, 51, 52, 53, 54, 55, 56, 57, 58, 59, 60]	87.50 ± 13.56 [65, 66, 67, 68, 69, 70, 71, 72, 73, 74, 75, 76, 77, 78, 79, 80, 81, 82, 83, 84, 85, 86, 87, 88, 89, 90, 91, 92, 93, 94, 95, 96, 97, 98, 99, 100]	0.690	**<0.001**	**0.010**
Role Physical (RP)	28.95 ± 21.87 [0–81.25]	16.67 ± 17.08 [0–43.75]	85.41 ± 16.92 [50, 51, 52, 53, 54, 55, 56, 57, 58, 59, 60, 61, 62, 63, 64, 65, 66, 67, 68, 69, 70, 71, 72, 73, 74, 75, 76, 77, 78, 79, 80, 81, 82, 83, 84, 85, 86, 87, 88, 89, 90, 91, 92, 93, 94, 95, 96, 97, 98, 99, 100]	0.652	**<0.001**	**0.002**
Bodily Pain (BP)	26.64 ± 18.39 [0–70]	18.75 ± 12.12 [0–35]	63.75 ± 16.32 [45, 46, 47, 48, 49, 50, 51, 52, 53, 54, 55, 56, 57, 58, 59, 60, 61, 62, 63, 64, 65, 66, 67, 68, 69, 70, 71, 72, 73, 74, 75, 76, 77, 78, 79, 80, 81, 82, 83, 84, 85, 86, 87, 88, 89, 90]	0.108	**0.002**	**<0.01**
General Health (GH)	29.68 ± 16.03 [0–65]	22.5 ± 19.69 [0–45]	69.58 ± 14.84 [35, 36, 37, 38, 39, 40, 41, 42, 43, 44, 45, 46, 47, 48, 49, 50, 51, 52, 53, 54, 55, 56, 57, 58, 59, 60, 61, 62, 63, 64, 65, 66, 67, 68, 69, 70, 71, 72, 73, 74, 75, 76, 77, 78, 79, 80, 81, 82, 83, 84, 85]	0.242	**<0.001**	**0.029**
Vitality (VT)	16.12 ± 15.83 [0–50]	10.42 ± 15.14 [0–37.5]	59.89 ± 10.47 [43.75–75]	0.182	**<0.001**	**0.003**
Social Functioning (SF)	35.20 ± 27.39 [0–87.5]	22.92 ± 18.40 [0–50]	88.54 ± 13.54 [62.5–100]	0.112	**0.002**	**<0.01**
Role Emotional (RE)	56.58 ± 37.12 [0–100]	43.06 ± 36.29 [0–83.33]	84.72 ± 20.04 [50, 51, 52, 53, 54, 55, 56, 57, 58, 59, 60, 61, 62, 63, 64, 65, 66, 67, 68, 69, 70, 71, 72, 73, 74, 75, 76, 77, 78, 79, 80, 81, 82, 83, 84, 85, 86, 87, 88, 89, 90, 91, 92, 93, 94, 95, 96, 97, 98, 99, 100]	0.305	0.174	0.060
Mental Health (MH)	47.24 ± 21.92 [5, 6, 7, 8, 9, 10, 11, 12, 13, 14, 15, 16, 17, 18, 19, 20, 21, 22, 23, 24, 25, 26, 27, 28, 29, 30, 31, 32, 33, 34, 35, 36, 37, 38, 39, 40, 41, 42, 43, 44, 45, 46, 47, 48, 49, 50, 51, 52, 53, 54, 55, 56, 57, 58, 59, 60, 61, 62, 63, 64, 65, 66, 67, 68, 69, 70, 71, 72, 73, 74, 75, 76, 77, 78, 79, 80, 81, 82, 83, 84, 85, 86, 87, 88, 89, 90]	40 ± 24.08 [10, 11, 12, 13, 14, 15, 16, 17, 18, 19, 20, 21, 22, 23, 24, 25, 26, 27, 28, 29, 30, 31, 32, 33, 34, 35, 36, 37, 38, 39, 40, 41, 42, 43, 44, 45, 46, 47, 48, 49, 50, 51, 52, 53, 54, 55, 56, 57, 58, 59, 60, 61, 62, 63, 64, 65, 66, 67, 68, 69, 70]	76.66 ± 18.25 [40, 41, 42, 43, 44, 45, 46, 47, 48, 49, 50, 51, 52, 53, 54, 55, 56, 57, 58, 59, 60, 61, 62, 63, 64, 65, 66, 67, 68, 69, 70, 71, 72, 73, 74, 75, 76, 77, 78, 79, 80, 81, 82, 83, 84, 85, 86, 87, 88, 89, 90, 91, 92, 93, 94, 95]	0.339	**0.008**	0.054

*p*-value (1) refers to the *p*-values obtained by comparing the complete FM cohort (*n* = 38) and the RNAseq FM subcohort (*n* = 6); *p*-value (2) refers to the *p*-values obtained by comparing the complete FM cohort (*n* = 38) and the and non-FM cohort (*n* = 12); and *p*-value (3) refers to the *p*-values obtained by comparing the RNAseq FM subcohort (*n* = 6) and non-FM cohort (*n* = 12). Statistically significant differences (*p* ≤ 0.05) appear bolded, and tendencies (*p* ≤ 0.1) are underlined.

**Table 2 ijms-25-09523-t002:** Participant baseline PPTs by studied cohort, as indicated. Patient tender point sensitivity assessment, as determined by triplicate measurements in lbf with an FDIX Force Gage, ForceOne algometer (Wagner Instruments, Greenwich, CT, USA) [12] at baseline.

	Total Cohort (*n* = 38)	RNAseq RNAseq Cohort (*n* = 6)	Non-FM Cohort (*n* = 12)			
Tender Points	Mean PPTs Pre- ± SD [Range]	Mean PPTs Pre- ± SD [Range]	Mean PPTs Pre- ± SD [Range]	*p*-Value (1)	*p*-Value (2)	*p*-Value (3)
Occiput right *	0.8062 ± 0.3771 [0.056–1.525]	0.908 ± 0.252 [0.348–1.135]	4.715 ± 1.457 [2.770–8.391]	0.555	**<0.001**	**<0.001**
Occiput left *	0.8606 ± 0.4029 [0.097–1.733]	0.931 ± 0.257 [0.367–1.270]	5.025 ± 2.056 [2.800–10.88]	0.696	**<0.001**	**0.001**
Trapezius right *	0.9371 ± 0.3521 [0.240–1.493]	0.985 ± 0.307 [0.398–1.937]	5.946 ± 1.739 [3.410–8.491]	0.052	**<0.001**	**0.002**
Trapezius left *	0.9757 ± 0.4149 [0.140–1.825]	1.126 ± 0.378 [0.398–1.883]	5.327 ± 1.336 [3.521–11.40]	0.246	**<0.001**	**0.003**
Supraspinatus right *	1.0115 ± 0.4311 [0.217–1.905]	1.202 ± 0.336 [0.550–1.968	6.506 ± 2.608 [3.384–13.38]	0.915	**<0.001**	**0.001**
Supraspinatus left *	1.0050 ± 0.4045 [0.177–1.838]	1.086 ± 0.438 [0.405–1.613]	6.687 ± 2.313 [3.010–12.15]	0.939	**<0.001**	**0.002**
Gluteal right *	1.3336 ± 0.6430 [0.158–2.780]	1.563 ± 0.654 [0.930–2.698]	8.657 ± 3.588 [4.651–15.37]	0.936	**<0.001**	**0.007**
Gluteal left *	1.3617 ± 0.6481 [0.207–2.670]	1.547 ± 0.598 [0.667–2.433]	9.13 ± 2.893 [4.120–14.34]	0.944	**<0.001**	**0.003**
Low cervical right *	0.4879 ± 0.2684 [0.000–1.172]	0.541 ± 0.134 [0.218–0.707]	3.028 ± 1.132 [1.291–5.611]	0.943	**<0.001**	**<0.001**
Low cervical left *	0.4732 ± 0.2347 [0.095–1.070]	0.434 ± 0.145 [0.183–0.625]	2.725 ± 1.361 [1.400–6.821]	0.332	**<0.001**	**<0.001**
Second rib right	0.7123 ± 0.4746 [0.152–2.665]	0.813 ± 0.322 [0.323–1.292]	5.281 ± 2.332 [3.060–11.15]	0.855	**<0.001**	**<0.001**
Second rib left	0.7036 ± 0.4530 [0.153–2.307]	0.806 ± 0.356 [0.238–1.338]	4.936 ± 2.771 [2.961–12.6]	0.948	**<0.001**	**<0.001**
Lateral epicondyle humerus right	0.8170 ± 0.4386 [0.080–1.830]	1.048 ± 0.331 [0.440–1.447]	5.868 ± 2.238 [2.722–11.52]	0.892	**<0.001**	**0.001**
Lateral epicondyle humerus left	0.8495 ± 0.3602 [0.298–1.823]	1.024 ± 0.311 [0.327–1.498]	5.719 ± 2.479 [3.133–12.34]	0.465	**<0.001**	**0.001**
Greater trochanter right	1.9234 ± 0.9089 [0.285–1.823]	2.077 ± 0.921 [0.548–3.362]	9.073 ± 2.266 [4.642–13.54]	0.133	**<0.001**	**0.002**
Greater trochander left	1.8306 ± 0.8524 [0.472–3.955]	1.928 ± 0.836 [0.793–3.195]	8.822 ± 2.875 [3.970–13.94]	0.532	**<0.001**	**0.003**
Knee right	1.1938 ± 0.6141 [0.263–2.505]	1.543 ± 0.618 [0.578–2.505]	8.766 ± 3.377 [3.511–16.80]	0.491	**<0.001**	**0.001**
Knee left	1.2958 ± 0.7296 [0.000–2.980]	1.637 ± 0.569 [0.542–1.930]	8.434 ± 3.007 [3.824–13.07]	0.977	**<0.001**	**0.002**

*p*-value (1) refers to the *p*-values obtained by comparing the complete FM cohort (n = 38) and the RNAseq FM subcohort (n = 6); *p*-value (2) refers to the *p*-values obtained by comparing the complete FM cohort (n = 38) and the and non-FM cohort (n = 12); and *p*-value (3) refers to the *p*-values obtained by comparing the RNAseq FM subcohort (n = 6) and non-FM cohort (n = 12). Statistically significant differences (*p* ≤ 0.05) appear bolded, and tendencies (*p* ≤ 0.1) are underlined. (*) Tender points in areas treated with manual therapy.

**Table 3 ijms-25-09523-t003:** DE-expressed genes with MT on PBMCs from FM patients, as determined by RNAseq.

Transcript	Post.Value	Pre.Value	FC (Post/Pre)	Log2FC	*p*-Value	Indiv.pval < 0.05	Gene_Name
UPREGULATED							
ENST00000361763	116.22	29.0968	3.994	1.997	0.0001	1.2.3.5.6	* **CD3E** *
ENST00000307271	20.6445	14.0276	1.472	0.557	0.00730	2.3.4	*GIMAP8*
ENST00000399220	76.4694	53.082	1.44	0.526	0.01760	1.2.3.4	* **CX3CR1** *
ENST00000296028	42.3176	29.9796	1.412	0.497	0.03415	1.4.5	*PPBP*
ENST00000304141	52.6702	39.3078	1.340	0.422	0.03320	1.4.5	*SDPR*
ENST00000367460	54.5804	41.3812	1.319	0.399	0.04980	1.2.4	*RGS18*
DOWNREGULATED							
ENST00000295924	10.7353	15.596	0.6883	−0.538	0.02835	2.3.4	*TIPARP*
ENST00000330871	51.7681	88.0488	0.588	−0.76624	0.00030	2.3.4.6	*SOCS3*
ENST00000288943	47.6877	90.7936	0.525	−0.92897	0.00160	2.4.5	*DUSP2*
ENST00000230990	3.1006	12.3126	0.251	−1.9895	0.0021	1.2.3.4.6	* **HBEGF** *
ENST00000307407	40.7346	164.584	0.2475	−2.0145	5.00 × 10^−5^	1.4.5.6	*CXCL8*
ENST00000369448	21.5097	43.1226	0.499	−1.0034	0.00005	1.3.5.6	*FAM46C*
ENST00000242480	3.61705	7.42934	0.487	−1.0384	0.01290	1.2.3.4.6	* **EGR2** *
ENST00000370626	1.7777	3.73306	0.048	−1.0703	0.00680	1.2.4	*AVPI1*
ENST00000377103	1.30687	2.91557	0.488	−1.1576	0.00045	1.2.6	*THBD*
ENST00000357949	3.02653	7.21969	0.419	−1.2542	0.00015	1.2.4	*SERTAD1*
ENST00000397806	139.994	354.549	0.395	−1.3406	0.00540	1.2.6	*HBA2*
ENST00000237305	13.9606	35.5195	0.393	−1.3472	0.03305	1.5.6	*SGK1*
ENST00000436139	3.12892	8.02824	0.390	−1.3594	0.00005	2.3.6	*RASGEF1B*
ENST00000379775	6.77581	17.804	0.3805	−1.3937	7.00 × 10^−4^	1.2.4	*PFKFB3*
ENST00000270162	0.63422	1.68599	0.004	−1.4105	0.00005	1.2.3.4.5.6	* **SIK1** *
ENST00000278175	10.4363	3.46589	0.301	−1.7316	0.01270	3.5.6	*ADM*
ENST00000508487	1.4087	6.71942	0.021	−2.2539	0.02150	1.5.6	*CXCL2*
ENST00000244869	0.492138	2.50443	0.002	−2.3473	0.02260	1.5.6	* **EREG** *

Numbers in the Indv.pval column indicate the FM participants who showed DE of the indicated genes with MT by RNAseq analysis (*p* < 0.05; FDR < 0.1), up- or downregulated, as indicated. FC: fold change. Genes selected for RT-qPCR validation appear bolded.

**Table 4 ijms-25-09523-t004:** Patient response to MT as evidenced by score differences in the standard, validated, FIQ, MFI, and SF-36 instruments [19,20,21,22] by the studied cohort. Significant differences (*p* ≤ 0.05) are bolded.

	Total Cohort (*n* = 38)				RNAseq RNAseq Cohort (*n* = 6)				Non-FM Cohort (*n* = 12)			
Questionnaire	Mean Pre- ± SD	Mean Post- ± SD	*p*-Value	Range	Mean Pre- ± SD	Mean Post- ± SD	*p*-Value	Range	Mean Pre- ± SD	Mean Post- ± SD	*p*-Value	Range
**FIQ**												
Total FIQ	72.62 ± 15.67	64.15 ± 18.25	**0.0334**	[41.08–96.51]	80.05 ± 17.26	68.71 ± 19.74	0.084	[46.12–92.09]	27.29 ± 13.28	26.97 ± 11.59	0.677	[5.64–49.64]
Function	5.16 ± 2.29	4.62 ± 2.43	0.3249	[0–9.24]	5.72 ± 2.11	5.225 ± 2.22	0.456	[3.3–8.25]	2.39 ± 0.50	2.42 ± 0.49	0.339	[1.98–3.63]
Overall	8.30 ± 2.23	6.74 ± 305	**0.0117**	[2.86–10.01]	8.10 ± 1.73	8.103 ± 2.811	1.000	[5.72–10.01]	9.17 ± 2.88	9.41 ± 1.42	0.674	[0–10.01]
Symptoms	4.59 ± 3.72	4.14 ± 3.32	0.2139	[0–10.01]	7.39 ± 3.77	4.05 ± 3.55	0.122	[0–10.01]	0.47 ± 1.11	0.47 ± 1.10	0.339	[0–2.86]
**MFI**												
General Fatigue	11.5 ± 1.6	11.7 ± 1.1	0.4383	[7, 8, 9, 10, 11, 12, 13, 14, 15, 16]	11 ± 1.79	11.83 ± 0.40	0.317	[8, 9, 10, 11, 12, 13]	11.41 ± 4.21	10.83 ± 3.56	**0.027**	[5, 6, 7, 8, 9, 10, 11, 12, 13, 14, 15, 16, 17]
Physical Fatigue	12.3 ± 1.2	12.4 ± 1.8	0.8124	[10, 11, 12, 13, 14, 15, 16]	12.83 ± 1.17	13.33 ± 1.96	0.490	[12, 13, 14, 15]	9.91 ± 3.75	9.66 ± 3.31	0.191	[6, 7, 8, 9, 10, 11, 12, 13, 14, 15, 16, 17]
Reduced Activity	12.1 ± 1.9	12.2 ± 2.3	0.8507	[6, 7, 8, 9, 10, 11, 12, 13, 14, 15, 16, 17, 18, 19]	11.5 ± 2.74	11.5 ± 1.37	1.000	[6, 7, 8, 9, 10, 11, 12, 13]	8.08 ± 2.87	8.16 ± 3.04	0.339	[4, 5, 6, 7, 8, 9, 10, 11, 12, 13, 14]
Reduced Motivation	10.6 ± 2.7	10.7 ± 2.6	0.8765	[4, 5, 6, 7, 8, 9, 10, 11, 12, 13, 14, 15, 16, 17, 18, 19]	10.83 ± 4.54	10.66 ± 2.58	0.872	[6, 7, 8, 9, 10, 11, 12, 13, 14, 15, 16, 17, 18, 19]	7.58 ± 3.14	7.5 ± 2.93	0.339	[4, 5, 6, 7, 8, 9, 10, 11, 12, 13, 14, 15]
Mental Fatigue	11.5 ± 1.8	11.9 ± 1.7	0.4786	[7, 8, 9, 10, 11, 12, 13, 14, 15]	11.5 ± 1.52	11 ± 1.09	0.597	[10, 11, 12, 13, 14]	7.08 ± 3.34	6.75 ± 3.01	0.104	[4, 5, 6, 7, 8, 9, 10, 11, 12, 13]
**SF-36**												
Physical Functioning (PF)	38.95 ± 17.48	41.46 ± 16.55	0.9486	[0–85]	37.5 ± 16.96	37.5 ± 14.74	1.000	[15, 16, 17, 18, 19, 20, 21, 22, 23, 24, 25, 26, 27, 28, 29, 30, 31, 32, 33, 34, 35, 36, 37, 38, 39, 40, 41, 42, 43, 44, 45, 46, 47, 48, 49, 50, 51, 52, 53, 54, 55, 56, 57, 58, 59, 60]	87.50 ± 13.56	87.50 ± 13.04	0.795	[65, 66, 67, 68, 69, 70, 71, 72, 73, 74, 75, 76, 77, 78, 79, 80, 81, 82, 83, 84, 85, 86, 87, 88, 89, 90, 91, 92, 93, 94, 95, 96, 97, 98, 99, 100]
Role Physical (RP)	28.95 ± 21.87	34.21 ± 25.24	0.7732	[0–81.25]	16.67 ± 17.08	37.5 ± 27.38	0.093	[0–43.75]	85.41 ± 16.92	85.93 ± 16.45	0.586	[50, 51, 52, 53, 54, 55, 56, 57, 58, 59, 60, 61, 62, 63, 64, 65, 66, 67, 68, 69, 70, 71, 72, 73, 74, 75, 76, 77, 78, 79, 80, 81, 82, 83, 84, 85, 86, 87, 88, 89, 90, 91, 92, 93, 94, 95, 96, 97, 98, 99, 100]
Bodily Pain (BP)	26.64 ± 18.39	36.45 ± 23.65	0.2341	[0–70]	18.75 ± 12.12	30 ± 19.74	0.112	[0–35]	63.75 ± 16.32	63.12 ± 16.72	0.339	[45, 46, 47, 48, 49, 50, 51, 52, 53, 54, 55, 56, 57, 58, 59, 60, 61, 62, 63, 64, 65, 66, 67, 68, 69, 70, 71, 72, 73, 74, 75, 76, 77, 78, 79, 80, 81, 82, 83, 84, 85, 86, 87, 88, 89, 90]
General Health (GH)	29.68 ± 16.03	27.76 ± 15.14	0.9010	[0–65]	22.5 ± 19.69	19.16 ± 14.28	0.286	[0–45]	69.58 ± 14.84	72.08 ± 12.14	0.191	[35, 36, 37, 38, 39, 40, 41, 42, 43, 44, 45, 46, 47, 48, 49, 50, 51, 52, 53, 54, 55, 56, 57, 58, 59, 60, 61, 62, 63, 64, 65, 66, 67, 68, 69, 70, 71, 72, 73, 74, 75, 76, 77, 78, 79, 80, 81, 82, 83, 84, 85]
Vitality (VT)	16.12 ± 15.83	20.53 ± 21.71	0.5948	[0–50]	10.42 ± 15.14	12.5 ± 15.81	0.576	[0–37.5]	59.89 ± 10.47	57.29 ± 14.05	0.096	[43.75–75]
Social Functioning (SF)	35.20 ± 27.39	46.91 ± 27.20	0.8543	[0–87.5]	22.92 ± 18.40	37.5 ± 27.38	0.135	[0–50]	88.54 ± 13.54	89.58 ± 13.93	0.586	[62.5–100]
Role Emotional (RE)	56.58 ± 37.12	53.51 ± 34.64	0.3037	[0–100]	43.06 ± 36.29	48.61 ± 30	0.444	[0–83.33]	84.72 ± 20.04	86.8 ± 17.57	0.082	[50, 51, 52, 53, 54, 55, 56, 57, 58, 59, 60, 61, 62, 63, 64, 65, 66, 67, 68, 69, 70, 71, 72, 73, 74, 75, 76, 77, 78, 79, 80, 81, 82, 83, 84, 85, 86, 87, 88, 89, 90, 91, 92, 93, 94, 95, 96, 97, 98, 99, 100]
Mental Health (MH)	47.24 ± 21.92	54.08 ± 22.08	0.9804	[5, 6, 7, 8, 9, 10, 11, 12, 13, 14, 15, 16, 17, 18, 19, 20, 21, 22, 23, 24, 25, 26, 27, 28, 29, 30, 31, 32, 33, 34, 35, 36, 37, 38, 39, 40, 41, 42, 43, 44, 45, 46, 47, 48, 49, 50, 51, 52, 53, 54, 55, 56, 57, 58, 59, 60, 61, 62, 63, 64, 65, 66, 67, 68, 69, 70, 71, 72, 73, 74, 75, 76, 77, 78, 79, 80, 81, 82, 83, 84, 85, 86, 87, 88, 89, 90]	40 ± 24.08	41.66 ± 22.94	0.846	[10, 11, 12, 13, 14, 15, 16, 17, 18, 19, 20, 21, 22, 23, 24, 25, 26, 27, 28, 29, 30, 31, 32, 33, 34, 35, 36, 37, 38, 39, 40, 41, 42, 43, 44, 45, 46, 47, 48, 49, 50, 51, 52, 53, 54, 55, 56, 57, 58, 59, 60, 61, 62, 63, 64, 65, 66, 67, 68, 69, 70]	76.66 ± 18.25	45.93 ± 10.17	**0.001**	[40, 41, 42, 43, 44, 45, 46, 47, 48, 49, 50, 51, 52, 53, 54, 55, 56, 57, 58, 59, 60, 61, 62, 63, 64, 65, 66, 67, 68, 69, 70, 71, 72, 73, 74, 75, 76, 77, 78, 79, 80, 81, 82, 83, 84, 85, 86, 87, 88, 89, 90, 91, 92, 93, 94, 95]

**Table 5 ijms-25-09523-t005:** Patient response to MT as evidenced by PPT differences by the studied cohort.

	Total Cohort (*n* = 38)				RNAseq RNAseq Cohort (*n* = 6)				Non-FM Cohort (*n* = 12)			
Tender Points	Mean PPTs Pre- ± SD	Mean PPTs Post- ± SD	*p*-Value	Range	Mean PPTs Pre- ± SD	Mean PPTs Post- ± SD	*p*-Value	Range	Mean PPTs Pre- ± SD	Mean PPTs Post- ± SD	*p*-Value	Range
Occiput right *	0.8062 ± 0.3771	0.8404 ± 0.3187	0.6732	[0.056–1.525]	0.908 ± 0.252	0.743 ± 0.307	0.457	[0.348–1.135]	4.715 ± 1.457	4.228 ± 1.179	0.104	[2.770–8.391]
Occiput left *	0.8606 ± 0.4029	0.8095 ± 0.3264	0.4646	[0.097–1.733]	0.931 ± 0.257	0.846 ± 0.351	0.640	[0.367–1.270]	5.025 ± 2.056	4.577 ± 1.872	0.167	[2.800–10.88]
Trapezius right *	0.9371 ± 0.3521	1.0741 ± 0.3597	0.0903	[0.240–1.493]	0.985 ± 0.307	1.173 ± 0.495	0.457	[0.398–1.937]	5.946 ± 1.739	5.172 ± 0.979	0.085	[3.410–8.491]
Trapezius left *	0.9757 ± 0.4149	0.9725 ± 0.6339	0.1890	[0.140–1.825]	1.126 ± 0.378	1.213 ± 0.487	0.735	[0.398–1.883]	5.327 ± 1.336	5.58 ± 2.189	0.587	[3.521–11.40]
Supraspinatus right *	1.0115 ± 0.4311	1.0115 ± 0.4139	0.4478	[0.217–1.905]	1.202 ± 0.336	1.416 ± 0.555	0.240	[0.550–1.968	6.506 ± 2.608	6.046 ± 1.845	0.285	[3.384–13.38]
Supraspinatus left *	1.0050 ± 0.4045	1.0050 ± 0.3470	0.3924	[0.177–1.838]	1.086 ± 0.438	1.113 ± 0.342	0.908	[0.405–1.613]	6.687 ± 2.313	6.081 ± 2.324	0.090	[3.010–12.15]
Gluteal right *	1.3336 ± 0.6430	1.3286 ± 0.5877	0.1359	[0.158–2.780]	1.563 ± 0.654	1.820 ± 0.692	0.524	[0.930–2.698]	8.657 ± 3.588	7.45 ± 1.697	0.156	[4.651–15.37]
Gluteal left *	1.3617 ± 0.6481	1.3550 ± 0.6178	0.2443	[0.207–2.670]	1.547 ± 0.598	1.470 ± 0.518	0.817	[0.667–2.433]	9.13 ± 2.893	8.214 ± 2.548	0.168	[4.120–14.34]
Low cervical right *	0.4879 ± 0.2684	0.4865 ± 0.1565	**0.0536**	[0.000–1.172]	0.541 ± 0.134	0.466 ± 0.186	0.438	[0.218–0.707]	3.028 ± 1.132	2.758 ± 1.183	0.333	[1.291–5.611]
Low cervical left *	0.4732 ± 0.2347	0.4774 ± 0.1235	**0.0197**	[0.095–1.070]	0.434 ± 0.145	0.418 ± 0.142	0.843	[0.183–0.625]	2.725 ± 1.361	2.641 ± 1.389	0.603	[1.400–6.821]
Second rib right	0.7123 ± 0.4746	0.7117 ± 0.2492	0.2971	[0.152–2.665]	0.813 ± 0.322	0.620 ± 0.344	0.340	[0.323–1.292]	5.281 ± 2.332	4.841 ± 2.308	**0.034**	[3.060–11.15]
Second rib left	0.7036 ± 0.4530	0.7098 ± 0.2691	0.8424	[0.153–2.307]	0.806 ± 0.356	0.664 ± 0.383	0.523	[0.238–1.338]	4.936 ± 2.771	4.853 ± 2.651	0.720	[2.961–12.6]
Lateral epicondyle humerus right	0.8170 ± 0.4386	0.8108 ± 0.2781	0.3622	[0.080–1.830]	1.048 ± 0.331	0.894 ± 0.315	0.428	[0.440–1.447]	5.868 ± 2.238	5.536 ± 2.122	0.324	[2.722–11.52]
Lateral epicondyle humerus left	0.8495 ± 0.3602	0.8495 ± 0.3054	0.1305	[0.298–1.823]	1.024 ± 0.311	0.738 ± 0.308	0.140	[0.327–1.498]	5.719 ± 2.479	5.335 ± 2.231	0.355	[3.133–12.34]
Greater trochanter right	1.9234 ± 0.9089	1.9144 ± 0.8721	0.3864	[0.285–1.823]	2.077 ± 0.921	1.858 ± 0.849	0.678	[0.548–3.362]	9.073 ± 2.266	8.155 ± 2.285	0.206	[4.642–13.54]
Greater trochander left	1.8306 ± 0.8524	1.7636 ± 0.8665	0.9034	[0.472–3.955]	1.928 ± 0.836	2.103 ± 0.973	0.746	[0.793–3.195]	8.822 ± 2.875	7.822 ± 2.274	0.263	[3.970–13.94]
Knee right	1.1938 ± 0.6141	1.1887 ± 0.3921	0.5056	[0.263–2.505]	1.543 ± 0.618	1.244 ± 0.468	0.368	[0.578–2.505]	8.766 ± 3.377	6.494 ± 3.411	**0.006**	[3.511–16.80]
Knee left	1.2958 ± 0.7296	1.3025 ± 0.4544	0.2608	[0.000–2.980]	1.637 ± 0.569	1.337 ± 0.428	0.327	[0.542–1.930]	8.434 ± 3.007	5.974 ± 2.355	**0.004**	[3.824–13.07]

(*) Tender points in areas treated with manual therapy; PPT (pressure point threshold); SD (standard deviation); pre- (pre-treatment); post- (post-treatment). Significant differences (*p* ≤ 0.05) are bolded.

## Data Availability

The raw data presented in this study have been deposited in Gene Expression Omnibus with ac-cession number GSE274134, https://www.ncbi.nlm.nih.gov/geo/query/acc.cgi?acc=GSE274134, accessed on 28 August 2024.

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
