# Peer review of "Manual Therapy Improves Fibromyalgia Symptoms by Downregulating SIK1"

_ijms, 2024, doi:10.3390/ijms25179523_

Round 1

Reviewer 1 Report

Comments and Suggestions for Authors

This manuscript discusses the contribution of a possible biomarker of response to manual therapy in FM. I find the idea intriguing, and your methodology and results provide interesting insights to understanding how FM patients respond to MT on a molecular level. I do have several comments for improvement.

Abstract: Line 21, please correct "other" to either "other symptoms" or "others". Similar typos across the manuscript should be fixed (e.g line 60 in the introduction, resulting "in" not "on")

Please mention the full name of every abbreviation you use at first use. For example, in the abstract you use PBMC, q PCR, etc..

Introduction: Lines 44-45: this sentence "nociplastic pain [1], caused by poorly understood mechanisms involving ongoing inflammation and
general tissue damage, rather than local nerve damage (neuropathic pain)"

I think this sentence is misleading. Nociplastic pain refers to pain that is centrally mediated by neurotransmitter mainly. Inflammation plays a role, but what do you mean by "general tissue damage"? how does nociplastic pain involve "tissue damage", when you actually need to rule out tissue damage that is nociceptive to diagnose nociplastic pain.

I would recommend revising the sentence to reflect the true nature of nociplastic pain.

The introduction lacks any background on the molecular and immune basis of your hypothesis. You should review the literature about PBMC transcriptomes and SIK in relation to FM.

Results:

Typo in line 93: diagnosis instead of diagnostic.

I realize the manuscript is already quite long, but I would have liked the authors to give a brief description of the tools they used, such as the FIQ and the MFI.

Lines 117 and 118: You state: while Gluteus, Trochanters and Knees were the most resistant to pain". In terms of PPT, certian points show high or low sensitivity. I am not sure the word "resistant to pain" could describe PPT. Please revise.

In your results in general, there are differences that are significant for the full FM cohort, but not for the 6 patients you choose for sequencing. This happens in the baseline FIQ and FMI parameters, as well as in the response to MT, and I wonder if it is still valid to refer to this group as a "representative" group. I think this needs to be addressed in your methodology, discussion and interpretation of results.

Discussion:

Some parts of the discussion are lengthy explanations of the role of SIK in FM, and this belongs in the introduction. The discussion should focus on explaining the relevance of your results, in relation to previous literature.

Overall, I think the work presented in this manuscript is very important and could guide efforts for future research.

Comments on the Quality of English Language

I would recommend thorough English editing for the whole manuscript.

Author Response

Response to Reviewer 1 Comments

1. Summary

Thank you very much for taking the time to review this manuscript. Please find the detailed responses below and the corresponding revisions/corrections highlighted/in track changes in the re-submitted files. [This is only a recommended summary. Please feel free to adjust it. We do suggest maintaining a neutral tone and thanking the reviewers for their contribution although the comments may be negative or off-target. If you disagree with the reviewer's comments please include any concerns you may have in the letter to the Academic Editor.]

2. Questions for General Evaluation

Reviewer’s Evaluation

Response and Revisions

Does the introduction provide sufficient background and include all relevant references?

Yes/Can be improved/Must be improved/Not applicable

[Please give your response if necessary. Or you can also give your corresponding response in the point-by-point response letter. The same as below]

Are all the cited references relevant to the research?

Yes/Can be improved/Must be improved/Not applicable

Please see point-by-point responses.

Is the research design appropriate?

Yes/Can be improved/Must be improved/Not applicable

Are the methods adequately described?

Yes/Can be improved/Must be improved/Not applicable

Are the results clearly presented?

Yes/Can be improved/Must be improved/Not applicable

Are the conclusions supported by the results?

Yes/Can be improved/Must be improved/Not applicable

3. Point-by-point response to Comments and Suggestions for Authors

Comments 1: This manuscript discusses the contribution of a possible biomarker of response to manual therapy in FM. I find the idea intriguing, and your methodology and results provide interesting insights to understanding how FM patients respond to MT on a molecular level. I do have several comments for improvement.

Response 1: thank you very much for the appreciation of our work.

Comments 2: Abstract: Line 21, please correct "other" to either "other symptoms" or "others". Similar typos across the manuscript should be fixed (e.g line 60 in the introduction, resulting "in" not "on")

Response 2: thank you. We have now corrected the indicated typos and reviewed the whole manuscript.

Comment 3: Please mention the full name of every abbreviation you use at first use. For example, in the abstract you use PBMC, q PCR, etc.

Response 3: thank you. We have now reviewed all abbreviations to ensure full name at first use in the Abstract and in the body of the manuscript.

Comment 4: Introduction: Lines 44-45: this sentence "nociplastic pain [1], caused by poorly understood mechanisms involving ongoing inflammation and
general tissue damage, rather than local nerve damage (neuropathic pain)"

I think this sentence is misleading. Nociplastic pain refers to pain that is centrally mediated by neurotransmitter mainly. Inflammation plays a role, but what do you mean by "general tissue damage"? how does nociplastic pain involve "tissue damage", when you actually need to rule out tissue damage that is nociceptive to diagnose nociplastic pain.

I would recommend revising the sentence to reflect the true nature of nociplastic pain.

Response 4: thank you for noticing. We have now removed the words “..involving ongoing inflammation and general tissue damage..” from the sentence to clarify this clinical key aspect of diagnosis.

Comment 5: The introduction lacks any background on the molecular and immune basis of your hypothesis. You should review the literature about PBMC transcriptomes and SIK in relation to FM.

Response 5: thank you for this suggestion. Although our starting hypothesis did not contemplate SIK1 as a gene “responding” to Manual Therapy (MT) treatment, but rather the possibility to monitor health improvement by molecular changes occurring in the immune system, we have added, following your very proper recommendation, a few lines in the Introduction to justify the adequacy to look at PBMC molecular profiles for responses on MT as suggested by previous studies.

Lines 61-76 adds this background justification with the corresponding bibliographic citations, as follows:

“Previously described effects of pressure-therapeutics, point at medium load pressure massage (4.5 N) maneuvers, including frequency and repetitions, to aid in muscle de-conditioning regrowth, typically lost during immobilization or in sedentary individuals, such as severely affected FM and/or CFS/ME patients [12], as described by Dupont-Vergesteegden´s group [13]. At the same time, and similarly to CBT (Cognitive Behavioral Therapy) and mindfulness, MT might engage patient’s mind into relaxation, boost happiness and promote immune, hormonal, and neurotransmitter responses [14,15].

Tissue reconditioning and patient symptom improvement are always preceded by molecular changes which often systemically spread through the body fluids, providing a low invasive opportunity to understand and monitor patient health-status through gene expression changes in the blood of patients. In particular, gene expression studies of FM PBMCs (Peripheral Blood Mononuclear Cells) have detected molecular differences co-inciding with immune cell activation [16], including hyperactivation of NK cells [17], positing the relevance of immunomodulatory therapeutics, such as MT. 

With the intention of elevating our knowledge of MT-mediated healthcare effects on FM,…”

Results:

Comment 6: Typo in line 93: diagnosis instead of diagnostic.

Response 6: thank you very much. It has been now fixed.

Comment 7: I realize the manuscript is already quite long, but I would have liked the authors to give a brief description of the tools they used, such as the FIQ and the MFI.

Response 7: thank you. This has been now added in the Introduction, so that readers can appreciate the overview of the working hypothesis and the methods applied to test it at an early stage of the reading: “This was achieved by studying the potential correlations between differential gene ex-pression (DE) in patient and control PBMCs (Peripheral Blood Mononuclear Cells) with MT and their potential correlation with changes in symptoms. DE was assessed by whole RNA sequencing (RNAseq) and Reverse Transcription followed by quantitative Poly-merase Chain Reaction (RT-qPCR) technologies, and changes in symptoms were meas-ured with the following validated instruments: FIQ (Fibromyalgia Impact Questionnaire) [19,20 ], MFI (Multi-Fatigue Inventory) [21], and SF-36 [22], towards assessing pain, fatigue and quality of life, respectively.”

Comment 8: Lines 117 and 118: You state: while Gluteus, Trochanters and Knees were the most resistant to pain". In terms of PPT, certian points show high or low sensitivity. I am not sure the word "resistant to pain" could describe PPT. Please revise.

Response 8: thank you. We have changed the term by “less sensitive”, but also added in the sentence what it is referred is “pressure-induced pain”.

Comment 9: In your results in general, there are differences that are significant for the full FM cohort, but not for the 6 patients you choose for sequencing. This happens in the baseline FIQ and FMI parameters, as well as in the response to MT, and I wonder if it is still valid to refer to this group as a "representative" group. I think this needs to be addressed in your methodology, discussion and interpretation of results.

Response 9: thank you. Although at baseline no major differences were detected in the subcohort that is called “representative” (Table 1, column Pvalue (1) shows no significant differences for any of the three instruments used, no differences for baseline PPTs (Table 2) except for one of the points Trapezius Right* would argue in favor, but we agree that some differences are detected in the response to therapy. While detection of significant differences in smaller groups (6 vs 38 individuals) may be harder to detect, we agree that may be more appropriate to define the 6-patient cohort as the “FM subcohort  or RNAseq cohort”. This has been adjusted now.

Discussion:

Comment 10: Some parts of the discussion are lengthy explanations of the role of SIK in FM, and this belongs in the introduction. The discussion should focus on explaining the relevance of your results, in relation to previous literature.

Response 10: thank you, but in this case, we respectfully disagree with presenting SIK1 in the introduction. First because, as mentioned in our response to Comment 5, our starting hypothesis did not contemplate SIK1 as a gene “responding” to Manual Therapy (MT) treatment. It is the result of the analysis that drives the attention to the downregulation of this gene. Secondly, because to the best of our knowledge differential expression of SIK1 has not been reported in FM. In addition, this study shows there are no basal differences of SIK1 between the full FM cohort (n=38) and the non-FM cohort (n=12) (please compare basal these basal levels on Figure 5). To get readers attention on this very important point, we have added the following sentence “However, SIK1 basal levels appear similar between the FM and the non-FM cohorts, but only significantly inhibited by MT in the FM group.” on lines 236 and following. With this scenario of a “novel finding”, we centered the discussion on highlighting the potential mechanisms that could explain our findings and corresponding future perspectives.

We have also added the following text in the Discussion, in support of the detailed potential downstream indirect actions of the detected inhibition of SIK1 (lines 374 and following):“Whether and how SIK1 changes affect FM immune transcriptome [16,17] will require further work. However, our finding that basal levels between the FM and non-FM cohorts appear similar in our RT-qPCR data, and the absence of a direct connection of SIK1 with FM phenotype, to the best of our knowledge, may indicate an indirect effect of SIK1 inhibition to mediate MT therapeutic action.”

And on lines 404-405: “..therefore offering a multitude of potential indirect mechanisms to exert MT therapeutic effects in FM.”

To offer potential indirect actions of SIK1 inhibition through innate immunity, circadian rhythm changes etc. for future work.

Comment 11: Overall, I think the work presented in this manuscript is very important and could guide efforts for future research.

Response 11: thank you very much for the appreciation of our work.

4. Response to Comments on the Quality of English Language

Point 1: Moderate editing

Response 1:  Reviewed  (in red)

5. Additional clarifications

[Here, mention any other clarifications you would like to provide to the journal editor/reviewer.]

Reviewer 2 Report

Comments and Suggestions for Authors

Comments and suggestions:

  1. The sample size, especially for the RNA sequencing portion (n=6), is quite small. This limits the statistical power and generalizability of the results.
  2. The study does not address potential confounding factors like medications participants may have been taking.
  3. The manual therapy protocol was brief (8 sessions over 4 weeks). A longer intervention period may have yielded more robust results.
  4. The mechanistic link between manual therapy and changes in gene expression in blood cells is not clearly explained.
  5. The statistical analysis does not appear to correct for multiple comparisons in all cases, which could lead to false positive results.
  6. Some of the correlations between gene expression changes and symptom improvements, while statistically significant, have relatively low R values, indicating weak associations.

Regarding Long COVID:

The authors do not specifically address or mention potential effects of Long COVID in this study. Given that:

  1. The study was conducted after the start of the COVID-19 pandemic
  2. There is overlap between some symptoms of fibromyalgia and Long COVID
  3. Long COVID could potentially impact gene expression patterns

It would have been appropriate for the authors to at least acknowledge this as a potential confounding factor or limitation of the study. They could have mentioned whether they screened participants for prior COVID-19 infection or current Long COVID symptoms. This omission represents a weakness in the study design and discussion, particularly given the timeframe in which the research was conducted.

Author Response

Response to Reviewer 2 Comments

1. Summary

Thank you very much for taking the time to review this manuscript. Please find the detailed responses below and the corresponding revisions/corrections highlighted/in track changes in the re-submitted files. [This is only a recommended summary. Please feel free to adjust it. We do suggest maintaining a neutral tone and thanking the reviewers for their contribution although the comments may be negative or off-target. If you disagree with the reviewer's comments please include any concerns you may have in the letter to the Academic Editor.]

2. Questions for General Evaluation

Reviewer’s Evaluation

Response and Revisions

Does the introduction provide sufficient background and include all relevant references?

Yes/Can be improved/Must be improved/Not applicable

response in the point-by-point

Are all the cited references relevant to the research?

Yes/Can be improved/Must be improved/Not applicable

Is the research design appropriate?

Yes/Can be improved/Must be improved/Not applicable

response in the point-by-point

Are the methods adequately described?

Yes/Can be improved/Must be improved/Not applicable

response in the point-by-point

Are the results clearly presented?

Yes/Can be improved/Must be improved/Not applicable

response in the point-by-point

Are the conclusions supported by the results?

Yes/Can be improved/Must be improved/Not applicable

response in the point-by-point

3. Point-by-point response to Comments and Suggestions for Authors

Comments 1: The sample size, especially for the RNA sequencing portion (n=6), is quite small. This limits the statistical power and generalizability of the results.

Response 1: we agree with the reviewer than the RNAseq cohort is rather small. However, it should be taken into consideration that the n=6 as participants means that the number of samples genome-wide sequenced were n=12 (pre/post), also that this doubles the standard experimental individual “triplicate”, and that the pre/post design favors comparison bias (same individual) in comparison to inter-individual designs. In addition, the RT-qPCR validated SIK1 findings in n=38 FM patients (n=76 samples). Being a completely novel exploratory approach, it seems justified to have a design in two phases: the “discovery phase” including a subset of samples that would need validation (“validation phase”) by a focused approach (RT-qPCR). We agree with the reviewer that the external validity of our novel finding is limited and that our pioneer findings await confirmation in additional extended cohorts and in response to other MT therapeutic protocols. 

Comments 2: The study does not address potential confounding factors like medications participants may have been taking.

Response 2: Response 2: thank you very much for this important point. This study was in fact using biobanked samples from a previous Clinical Trial that described more in detail patient inclusion criteria. However, given the relevance of potential interference of medication, we have now added a sentence to highlight that potential confounding factors were controlled as much as possible by a careful design. The sentence added on lines 444 and following is: “Participants could not be enrolled in pharmacological CTs or receive additional physiotherapy treatment while participating in this study and agreed to withdraw medication 12 h before blood draws.”

Comment 3: The manual therapy protocol was brief (8 sessions over 4 weeks). A longer

intervention period may have yielded more robust results.

Response 3: we agree with the reviewer that longer treatments may provide even more benefits. However, the previous clinical trial that generated the FM samples that were analyzed already showed that this 8-week protocol provided benefits to FM patients (citation 12, now renumbered as 18; Falaguera-Vera et al., IJERPH 2020).

Comment 4: The mechanistic link between manual therapy and changes in gene expression in blood cells is not clearly explained.

Response 4: thank you for noticing. We have now added lines 61-76 to clarify this important background with the corresponding citations, as follows: “Previously described effects of pressure-therapeutics, point at medium load pressure massage (4.5 N) maneuvers, including frequency and repetitions, to aid in muscle de-conditioning regrowth, typically lost during immobilization or in sedentary individuals, such as severely affected FM and/or CFS/ME patients [12, as described by Dupont-Vergesteegden´s group [13]. At the same time, and similarly to CBT (Cognitive Behavioral Therapy) and mindfulness, MT might engage patient’s mind into relaxation, boost happiness and promote immune, hormonal, and neurotransmitter responses [14,15].

Tissue reconditioning and patient symptom improvement are always preceded by molecular changes which often systemically spread through the body fluids, providing a low invasive opportunity to understand and monitor patient health-status through gene expression changes in the blood of patients. In particular, gene expression studies of FM PBMCs (Peripheral Blood Mononuclear Cells) have detected molecular differences co-inciding with immune cell activation [16], including hyperactivation of NK cells [17], positing the relevance of immunomodulatory therapeutics, such as MT. 

With the intention of elevating our knowledge of MT-mediated healthcare effects on FM,…”

Comment 5: The statistical analysis does not appear to correct for multiple comparisons in all cases, which could lead to false positive results.

Response 5: we agree with the reviewer that validation statistics must be as restrictive as possible to minimize false positives. While exploratory phases with few participants may leave biologically relevant players out of consideration, we have validated the performance of SIK1 in our cohort with very stringent low p-values, supporting robustness of our main findings.

Comment 6: Some of the correlations between gene expression changes and symptom improvements, while statistically significant, have relatively low R values, indicating weak associations.

Response 6: we agree with the reviewer that the R value of SIK1 levels with bodily pain seems low (R=-0.48), but the improvement of several symptom domains fall within the moderate range (R=0.5-0.7) (Fig. 6). In the case of bodily pain, it should be highlighted that after considering ME/CFS co-diagnosis, meaning that the “n” was reduced to half of the cases (n=19), the R value increased to top moderate R= -0.66 only in the FM subgroup  (Fig. 8), importantly indicating that SIK1 must be a specific therapeutic target in this subset of FM patients. MT being a non-pharmacological therapeutic approach is not expected to become a cure, but rather adjuvant therapy. This view should also be considered when studying healing in FM.

Regarding Long COVID:

The authors do not specifically address or mention potential effects of Long COVID in this study. Given that:

Comment 7: The study was conducted after the start of the COVID-19 pandemic. There is overlap between some symptoms of fibromyalgia and Long COVID. Long COVID could potentially impact gene expression patterns. It would have been appropriate for the authors to at least acknowledge this as a potential confounding factor or limitation of the study. They could have mentioned whether they screened participants for prior COVID-19 infection or current Long COVID symptoms. This omission represents a weakness in the study design and discussion, particularly given the timeframe in which the research was conducted.

Response 7: we agree with the reviewer that Long COVID presents symptoms that overlap with FM and ME/CFS, could impact immune gene expression patterns, and contribute as confounding factors. However, the FM samples analyzed were biobanked samples from a registered Clinical trial (NCT04174300) that was completed just before SARS-CoV-2 pandemia. As this is an important point to be highlighted, we have added this information in the 2.1.1. Results section, lines 92 and following: “Due to the frequent symptom overlap with post-COVID-19 condition (popularly known as Long COVID) [23] it should be highlighted that the FM cohort studied is pre-pandemic (NCT04174300 completed before 03/2020).”

Study Details | Molecular Response to Custom Manual Physiotherapy Treatment of Fibromyalgia & Chronic Fatigue Syndrome (CFS) | ClinicalTrials.gov

4. Response to Comments on the Quality of English Language

Point 1:

Response 1:    (in red)

5. Additional clarifications

Round 2

Reviewer 2 Report

Comments and Suggestions for Authors

The authors responded to my comments very well. Thank you.